# Motor response vigour and visual fixation patterns reflect subjective valuation during intertemporal choice

Elke Smith 📷 *, Jan Peters 📷

Department of Psychology, Biological Psychology, University of Cologne, Cologne, Germany

* e.smith@uni-koeln.de

**Data Availability Statement:** The data cannot be shared publicly, since the participants did not provide consent to public data access. The raw data and the data underlying the tables and figures are available from RADAR at https://www.radar-

## Abstract

Value-based decision-making is of central interest in cognitive neuroscience and psychology, as well as in the context of neuropsychiatric disorders characterised by decision-making impairments. Studies examining (neuro-)computational mechanisms underlying choice behaviour typically focus on participants' decisions. However, there is increasing evidence that option valuation might also be reflected in motor response vigour and eye movements, implicit measures of subjective utility. To examine motor response vigour and visual fixation correlates of option valuation in intertemporal choice, we set up a task where the participants selected an option by pressing a grip force transducer, simultaneously tracking fixation shifts between options. As outlined in our preregistration (https://osf.io/k6jct), we used hierarchical Bayesian parameter estimation to model the choices assuming hyperbolic discounting, compared variants of the softmax and drift diffusion model, and assessed the relationship between response vigour and the estimated model parameters. The behavioural data were best explained by a drift diffusion model specifying a non-linear scaling of the drift rate by the subjective value differences. Replicating previous findings, we found a magnitude effect for temporal discounting, such that higher rewards were discounted less. This magnitude effect was further reflected in motor response vigour, such that stronger forces were exerted in the *high* vs. the *low* magnitude condition. Bayesian hierarchical linear regression further revealed higher grip forces, faster response times and a lower number of fixation shifts for trials with higher subjective value differences. An exploratory analysis revealed that subjective value sums across options showed an even more pronounced association with trial-wise grip force amplitudes. Our data suggest that subjective utility or implicit valuation is reflected in motor response vigour and visual fixation patterns during intertemporal choice. Taking into account response vigour might thus provide deeper insight into decision-making, reward valuation and maladaptive changes in these processes, e.g. in the context of neuropsychiatric disorders.

service.eu/radar/en/dataset/PtHBqKheaYugZaWh
(DOI: 10.57743/612). Access is granted to
researchers for scientific purposes only. All model
code is openly available on the Open Science
Framework at https://osf.io/agqsp/ (DOI: 10.17605/
OSF.IO/AGQSP).

**Funding:** This work was funded by the DFG (grant
PE1627/5-1, awarded to J. P., https://www.dfg.de/
). The funders had no role in study design, data
collection and analysis, decision to publish, or
preparation of the manuscript.

**Competing interests:** The authors have declared
that no competing interests exist.

## Author summary

Value-based decision-making is a process of particular interest in cognitive neuroscience. Impairments in decision-making are hallmarks of several neuropsychiatric disorders, and specifically temporal discounting, i.e. the devaluation of future rewards, is discussed as a potential marker for psychiatric disorders. Subjective utility during value-based decision-making is commonly inferred on the basis of participants' choices only. Here were investigated if motor response vigour and visual fixation patterns may serve as additional and implicit measures of subjective utility during intertemporal choice. We demonstrate that motor response vigour and visual fixation patterns are related to the reward magnitudes and subjective value differences between options. Further, we show that the choice and response time data are well accounted for by a drift diffusion model including a non-linear scaling of the drift rate by the subjective option value differences. Our results suggest that measures of motor response vigour and visual fixation patterns may provide further insight on valuation during decision-making when combined with choice and response time data.

## Introduction

Motivation entails the willingness to perform effortful actions in order to obtain rewards. Individuals normally adapt the level of effort expended to the expected utility of a reward. An adequate adjustment of effort to expected utility is crucial to ensure reward receipt, whilst avoiding unnecessary energy expenditure. Whether a reward is worth a given effort depends on its expected (subjective) utilty. The expected utility of a reward does not equal its utility in an absolute sense, but is contingent upon both intraindividual and external factors. For instance, rewards that are temporally more distant are typically devaluated, resulting in a preference for smaller, but sooner rewards, over larger, but later rewards, a process known as temporal discounting [1, 2].

The degree of discounting delayed rewards has been linked to a range of harmful behaviours and psychiatric conditions, including impulsivity, substance abuse and addiction (for a review, see [3]). For instance, individuals suffering from substance use disorders appear to be biased towards choosing immediate compared to delayed, but larger, rewards [4, 5].

Key brain circuits involved in value-based decision-making include the medial prefrontal cortex and striatum. Here, brain activity correlates with subjective value in a variety of tasks, such as valuation of goods and intertemporal choice [6–8]. The devaluation of rewards by both cognitive and physical effort appears to be associated with BOLD activation in mostly overlapping neural structures [9].

It is well established that midbrain dopaminergic neurons play a central role in decision-making and reward processing [10, 11]. Direct evidence for the involvement of dopamine in effort-based decision-making comes from studies in patients with Parkinson's disease (PD) and from pharmacological studies manipulating dopamine transmission. In patients with PD, effort-based decision-making appears to be disrupted—they have been found to exert less force for rewards compared to healthy controls, and to exert less force when being off compared to on their dopaminergic medication [12, 13]. In turn, pharmacological enhancement of dopamine transmission via levodopa in healthy participants increased the force levels exerted to obtain high vs. low rewards. Following debriefing, none of the participants reported to exert higher forces to obtain high rewards [14], suggesting that the behaviour reflects an implicit motivational process [14].

Pessiglione and colleagues [15] found that participants exert more force to obtain higher rewards even in cases where the rewards have only been presented subliminally. Also, across different social contexts (collaborative and competitive), force production was strongly related to subjective utility, and increased with absolute monetary value [16]. Further, subjective utility in value-based decision-making is reflected in eye movement vigour [17]. For instance, as participants approach their decision, eye movement vigour (i.e. peak velocity of a saccade as a function of amplitude) becomes greater for the preferred reward option, and the difference in eye movement vigour is closely linked to the difference in assigned subjective values of the options [18].

While value-based decision-making is a complex process requiring information integration, value computation and comparison, in most experimental settings, the process of evaluating a reward's utility is often inferred from the participants' choices only. However, from the above findings it appears that measures of response vigour may provide additional insights into motivation and value-based decision-making rather than measures of choice behaviour alone. In the present study, we therefore investigated if measures of response vigour, specifically handgrip force applied during choice selection, and visual fixation patterns may serve as implicit measure of outcome utility and decision conflict during intertemporal choice. There are well-established models to describe subjective valuation during intertemporal choice, allowing for a well-grounded modelling of the relationship between response vigour and subjective utility [19–21].

In contrast to the incentive force task used by Pessiglione and colleagues [15, 16], where the force applied was directly related to the payout and visually fed back to the participants, we captured implicit motivational processes by keeping the amount of force produced hidden from the participants and unrelated to the payout. Besides being instrumental in obtaining a reward, the allocation of effort may also be a correlate of the underlying evaluation process. We further included an experimental manipulation known to substantially affect reward valuation during temporal discounting, the magnitude effect [22, 23]. This effect describes the reduction in discount rates that occurs during intertemporal choice for increasing reward amounts, and we explored whether this effect is also reflected in the handgrip response.

Models of value-based decision-making, including temporal discounting, typically implement action selection using the softmax function [24]. We extend this approach by jointly modelling the choices and response times (RTs) with the drift diffusion model (DDM) [25], a form of sequential sampling model for two-alternative forced choice tasks. The drift diffusion model assumes that choices are driven by a noisy accumulation process, which terminates as soon as the level of accumulated evidence has reached one of two response boundaries. The model's strength lies in the incorporation of both choices and RTs in the model estimation. It has proven to be a useful model in explaining choice behaviour and RTs during value-based decision-making in our and others' prior work [19, 26–29].

We analysed the relationship between the subjective value differences as derived from the estimated drift diffusion model parameters and the force applied and fixation shifts during response selection. Further, we assessed the relationship between decision conflict, motor response vigour and visual fixation patterns. As outlined in the preregistration of our study (https://osf.io/k6jct), we tested the following hypotheses:

(i). Delay influences reward evaluation: Participants show a tendency to devaluate rewards that are temporally distant (temporal discounting).

(ii). Differences in subjective utility modulate response times and grip force: Faster response times and stronger effort (handgrip force) in trials with high subjective value differences.

(iii). Decision conflict is reflected in motor response vigour and visual fixation patterns: Longer deliberation (response time), less vigour (grip force) and more frequent fixation shifts between the options during high-conflict decisions (choice options with similar subjective value).

(iv). Higher rewards are discounted less and elicit more effort: Lower discount rates, faster response times, and greater motor response vigour (grip force) for larger rewards (between-conditions magnitude effect).

## Materials and methods

### Ethics statement

The study was approved by the local institutional review board (Ethics Committee of the Medical Faculty of the University of Cologne) and all participants provided informed written consent.

### Sample

Based on the effect sizes of previous studies reporting a magnitude effect for temporal discounting and handgrip force, respectively [15, 22, 23], a power analysis yielded a sample size of $N = 20$ (effect size Cohen's $d_z = 1.1698$ and $d_z = -0.7481$, respectively, $\alpha$ error probability = .05, power = .95, one-tailed paired $t$-test). We doubled the sample size and tested 42 participants in total. As two participants had to be excluded due to technical issues, the final sample consisted of $N = 40$ participants (30 women, 34 right-handed, 39 with German Abitur, 1 with German Mittlere Reife or GCSE), aged 18 to 39 ($M = 23.95$, $SD = 4.90$).

The participants were recruited through university bulletins, mailing lists and by word-of-mouth recommendation. Eligibility criteria included normal or corrected-to-normal vision and German as first language (or profound German language skills). Participants with strongly impaired vision, strabismus and psychiatric disorders were excluded.

### Task

The study was implemented as one-group, repeated-measures within-subject design, including two conditions. The participants performed 192 trials of an intertemporal choice task, whereby they had to choose between smaller-but-sooner (SS) and larger-but-later (LL) rewards. On one half of the trials, the SS reward was lower (10 €, *low* condition), and on the other half the SS reward was higher (20 €, *high* condition). The SS reward was always available immediately, while the LL reward consisted of combinations of sixteen ratios of the SS reward value [1.03 1.05 1.10 1.15 1.20 1.25 1.35 1.45 1.50 1.70 1.90 2.20 2.50 2.90 3.30 3.80] and six delay periods in days [1 7 13 31 58 122]. The order of the trials and the assignment of the options to the left and right side of the screen were presented in randomised order. The participants were financially reimbursed for participation and additionally received the payout from one randomly selected trial (restricted to maximum 40 €).

### Experimental setup

The measurements took place at the Psychology Department of the University of Cologne. During testing, the participants were seated in a dimly lit, electrically and acoustically shielded room, with their head placed in a chinrest. Prior to the experiment, they were instructed to press the handgrip with maximal force three times in succession with their dominant hand.

The procedure was disguised as calibration procedure. Following that, the participants were instructed that the level of force exerted is irrelevant to the task structure.

After presenting both options, one of the two options could be preselected through visual fixation. For this purpose, we used an eyetracking system (SensoMotoric Instruments, Model: RED 500, sampling rate: 500 Hz) to track the fixation patterns and highlight the currently fixated reward option in real-time. For highlighting the fixated option, the corresponding screen areas were defined as follows: Left area $<= \frac{1}{10}$ screen pixels of x-coordinate x 4, middle area $>= \frac{1}{10}$ screen pixels of x-coordinate x 4 and $<= \frac{1}{10}$ screen pixels of x-coordinate x 6, right area $>= \frac{1}{10}$ screen pixels of x-coordinate x 6.

The responses were logged using a hand dynamometer measuring grip force (BIOPAC Systems, Inc., Model: TSD121C, isometric range: 0–100 kgf). The force threshold to register a choice was set to 0.70 kgf. The threshold was determined in pilot measurements in such a way that false positive signals, caused by holding and slightly moving the force transducer, were avoided, while at the same time ensuring that no effort was required for a response. There was no response time limit. After having preselected an option through visual fixation, participants could still deliberate and decide for the other option as long as the force transducer had not been pressed. The measured variables included the participants' choices, response times, fixation shift patterns and handgrip force applied during response selection, as well as their maximum handgrip force.

## Data analyses

**Preprocessing.** All logfiles were checked for stereotypic response patterns (exclusively SS or LL choices), none were found. Choice patterns consisting of exclusively SS or LL choices may indicate that the participants proceeded heuristically rather than including values and delay periods in their reasoning. Valid response times are physiologically limited to a lower bound of around 100 to 200 ms [30, 31]. Since even implausibly fast outlier trials must be assigned a probability density $> 0$, modelled response time distributions for a given participant are shifted towards zero as much as required to accommodate for such response times. This may lead to poor model fits at the level of individual participants, and consequently may also impact on the fits of hierarchical models. Therefore, we excluded trials with response times below 200 ms. Further, we excluded trials with response times $> 10$ s. The participants were instructed that there was no time pressure for the decision, but that they should decide according to their gut feeling and not think long about the decision. Since the task was comparatively simple, long reaction times likely reflect a lack of attention rather than the process of interest. Finally, we excluded trials with maximum grip force values falling below the threshold for logging a response (technical issue with faulty signal on parallel port). In total, 139 trials (1.81% of trials) from 26 participants were excluded. The grip force data were further baseline-corrected to zero, normalised to each participant's maximal voluntary contraction (MVC, greatest force exerted over three contractions), and smoothed with a moving average of 50 samples.

**Computational modelling of behaviour. Temporal discounting model**. Ensuing from previous research on the effects of immediacy vs. delay on choice behaviour, we assume temporal discounting to be hyperbolic [32, 33]. We quantified the discount rates using a model-based approach of hyperbolic discounting. To capture the choice behaviour in both conditions within a single model, we fitted a single subject-specific discount rate parameter $k$ (estimated in logarithmic space), modelling the discount rate in the *low* condition, plus a subject-specific parameter $s$, modelling the change in the discount rate from the *low* compared to

the *high* condition.

$$SV(LL)_t = \frac{A_t}{1 + e^{k+I_t * s_k} * IRI_t} \tag{1}$$

Here, *SV* is the subjective (discounted) value of the delayed reward and *A* is the amount of the LL reward on trial *t*. *K* is the (subject-specific) discount rate for the *low* condition (in logarithmic space), *s* is a (subject-specific) shift in *log(k)* from the *low* to *high* condition, *I* is a condition indicator variable (zero for *low* trials, one for *high* trials), and *IRI* is the inter-reward-interval.

**Softmax choice rule**. The softmax action selection rule is a commonly used choice rule for value-based decision making and reinforcement learning, applied in our own and others' previous work [24, 34, 35]. Here we applied this model as a baseline or reference model. The softmax choice rule models the probability of choosing the LL reward on trial *t* as

$$P(LL)_t = \frac{e^{\beta * SV(LL_t)}}{e^{\beta * SV(LL_t)} + e^{\beta * SV(SS_t)}} \tag{2}$$

*SV* is the subjective value of the LL option, and $\beta$ is an inverse temperature parameter, describing the stochasticity of the choices (for $\beta = 0$ the choices are random, while as $\beta$ increases, the choices become increasingly dependent on the values of the options).

**Drift diffusion model**. We further modelled the participants' choices using the *drift diffusion model* (DDM), whereby the softmax choice rule is replaced by the drift diffusion choice rule. For the boundary definitions of the DDM, we applied stimulus coding, with the lower boundary defined as choosing the SS reward, and the upper boundary defined as choosing the LL reward. For this purpose, choices towards the lower boundary were multiplied by -1. When using absolute RT cut-offs, single fast trials force model parameters to adapt these trials und hence lead to a poor model fit at the single-subject level [19]. We therefore excluded each participant's slowest and fastest 2.5% trials from the analysis. The response time on trial *t* is distributed following the *Wiener first passage time* (WFPT):

$$RT_t \sim wfpt(\alpha, \tau, z, \upsilon) \tag{3}$$

The parameter $\alpha$ reflects the boundary separation (modelling a speed-accuracy trade-off), $\tau$ is the non-decision time (modelling processing time unrelated to the decision process), $\upsilon$ is the drift rate (modelling the rate of evidence accumulation), and $z$ is the starting-point bias (modelling a bias towards one of the boundaries). Using the JAGS Wiener module [36], $z$ may range between 0 and 1, whereby $z = .5$ indicates no bias in either direction, $z < .05$ indicates a bias towards the lower boundary (SS option), and $z > .05$ indicates a bias towards the upper boundary (LL option). First, we fitted a *null* model (DDM$_0$) without value modulation. This model comprises four parameters ($\alpha$, $\tau$, $z$, and $\upsilon$), which are constant across trials for each participant. To connect the drift diffusion model with the valuation model (see Eq 1), we implemented two further models comprising a function which links the trial-by-trial variability in the drift rate $\upsilon$ to the value differences. First, we realised a *linear* model (DDM$_{lin}$), following Pedersen, Frank, and Biele [37]:

$$\upsilon_t = \upsilon_{coeff} * (SV(LL_t) - SV(SS_t)) \tag{4}$$

The parameter $\upsilon_{coeff}$ maps the value differences onto the drift rate $\upsilon$ and transforms these differences to the proper scale of the DDM [37]. As a last step, we implemented a *sigmoid* model (DDM$_{sig}$), entailing a non-linear transformation of the scaled value differences with an *S*-shaped function as proposed by Fontanesi, Gluth, Spektor, and Rieskamp [26], where *S* is a

sigmoid function centred at zero with slope $m$ and asymptote $\pm v_{max}$:

$$v_t = S(v_{coeff} * (SV(LL_t) - SV(SS_t)))$$ (5)

$$S(m) = \frac{2 * v_{max}}{1 + e^{-m}} - v_{max}$$ (6)

Ensuing from this model, we also realised a *shift* model (DDM$_{shift}$), including the parameters $s_\alpha$, $s_\tau$, $s_z$, $s_v$, $s_{v_{coeff}}$, and $s_{v_{max}}$ to model changes in the parameter distributions from the *low* to *high* condition:

$$RT_t \sim wfpt(\alpha + I_t * s_\alpha, \tau + I_t * s_\tau, z + I_t * s_z, v + I_t * s_v)$$ (7)

$$v_t = S(v_{coeff} + I_t * s_{v_{coeff}} * (SV(LL_t) - SV(SS_t)))$$ (8)

$$S(m) = \frac{2 * (v_{max} + I_t * s_{v_{max}})}{1 + e^{-m}} - (v_{max} + I_t * s_{v_{max}})$$ (9)

Since the drift rate depends on the absolute magnitudes of the values, which, in turn differ between the *low* and *high* condition, condition effects are somewhat difficult to interpret. Extending the modelling as set out in the preregistration plan, we therefore further compared the drift diffusion models using absolute vs. normalised values (normalised by the maximum value of the LL reward per magnitude condition).

**Decision conflict**. To assess the hypothesised relationship between decision conflict, motor response vigour and visual fixation patterns, we considered two different operationalisations of decision conflict, based on (i) the choice probability from the softmax choice rule and (ii) the trial-wise drift rate as derived from the DDM. For decision conflict based on the softmax model, we defined decision conflict from 1 (low conflict) to 5 (high conflict), with a probability of 0.5 of choosing the LL reward as maximum conflict. To provide a common scaling from low to high conflict, probabilities $> 0.5$ were 'flipped' $(1 - p)$, implying that for instance a probability of 0.1 and 0.9, respectively, of choosing the LL reward represent an equally low decision conflict (choose SS with high probability, and choose LL with high probability, respectively). The data were grouped into five bins using MATLAB's `discretize` and `accumarray` function (choice probabilities between 0 and 0.1 assigned to bin 1, choice probabilities between 0.4 and 0.5 assigned to bin 5, etc.).

Further, extending our planned analyses, we assessed the relationship between motor response vigour, visual fixation patterns and the subjective value differences and sums, respectively, based on the estimated parameters of the drift diffusion model (using absolute subjective values).

**Motor response vigour and fixation shifts.** **Motor response vigour (grip response).** To examine the relationship between the characteristics of the handgrip response and the choice behaviour and estimated model parameters (subjective value differences, choice probabilities and decision conflict), we modelled the handgrip response on individual trials with a Gaussian function of the form

$$f(x) = ae^{-\left(\frac{x-b}{c}\right)^2} + h$$ (10)

using MATLAB's `fit` function, where the coefficient $a$ is the amplitude (height of peak), $b$ the centroid (centre of peak), $c$ the width (width of peak) and $h$ is a constant (to model offsets from zero). The handgrip data were fitted trial-wise per participant. To test for a magnitude effect in the grip force response, we used frequentist significance tests (one-tailed for

amplitude and centroid, see section Introduction, hypothesis iv, significance threshold set at .05, not corrected for multiple comparisons).

**Fixation shifts**. Using the eye tracking data, we assessed the relationship between the frequency of fixation shifts between the choice options and the associated decision conflict (see section Decision conflict). We defined fixation shifts as the number of switches between the left and right option (skipping middle fixations, see section Task).

**Effects of conflict, value difference and value sum.** To assess the effects of conflict, we regressed motor response vigour (single-trial Gaussian grip force model parameters) and the number of fixation shifts onto the response conflict measures. We fitted a hierarchical Bayesian linear regression of the form

$$y_t = \alpha + \beta_1 * a_t + \beta_2 * b_t + \beta_3 * c_t + \beta_4 * d_t \tag{11}$$

where $y$ is the conflict on trial $t$, operationalised either (1) based on the choice probabilities from the softmax model (see section Decision conflict), (2) as the trial-wise drift rate, based on the estimated parameters of the best fitting drift diffusion model, or (3) as the value difference between the (discounted) LL and SS reward on trial $t$, based on the estimated subject-specific $k$ parameters of the DDM (see Eq 1).

Since we observed no relationship between motor response vigour, number of fixation shifts and conflict, neither for choice probability (softmax model) nor trial-wise drift rate (DDM), we extended our analyses plan and also regressed motor response vigour and number of fixation shifts on the (absolute) subjective value differences. We reasoned that this might be attributable to the fact that both predictors are insensitive to increasingly higher value differences: in the softmax model, these are mapped to a conflict of 0, whereas in the DDM these are mapped to a maximum drift rate of $v_{max}$. As we observed a magnitude effect for grip force amplitude, we carried out a further exploratory analysis to test whether the total value (sum across options) would likewise show an association with response vigour. To this end, we regressed grip force and the number of fixation shifts onto the sum of the LL and SS option amounts (see Section 5 and Fig F in S1 Text of the supplementary material) and onto the sum of the subjective LL and SS option values, based on the discount rates estimated from the drift diffusion model. These models were not preregistered.

The estimated grip force parameters $a$, $b$, and $c$, and the number of fixation shifts were within-subjects $z$-standardised before entering the regression. The parameter $d$ corresponds to the absolute number of fixation shifts between the options. Since we excluded each participant's slowest and fastest 2.5% of trials within the scope of the drift diffusion model (see section Drift diffusion model), the respective trials were likewise removed from the grip force and gaze data.

We report Bayes factors (BFs) for directional effects [38] for the $\beta$—hyperparameters, via kernel density estimation in MATLAB (The MathWorks, Inc., version R2019a). The Bayes factors are defined as the ratio of the integral of the posterior distribution from—$\infty$ to 0 versus the integral from 0 to $\infty$. We consider BFs between 1 and 3 as anecdotal evidence, BFs between 3 and 10 as moderate evidence, BFs between 10 and 30 as strong evidence, BFs between 30 and 100 as very strong evidence, and BFs above 100 as extreme evidence for the H1. The inverse of these values reflect the corresponding evidence for the H0 [39, 40]. We further report the posterior highest density intervals (HDI) along with the regions of practical equivalence (ROPE, limits for $\beta$ = ±0.05 as for standardised variables) [41] for the posterior distributions of the regression coefficients.

**Parameter estimation and model comparison.** The parameter distributions of the softmax, drift diffusion and regression models were estimated through Markov chain Monte

Carlo (MCMC) simulation as implemented in JAGS [42, version 4.3.0], using MATLAB (The MathWorks, Inc., version R2019a) and the MATJAGS inferface for JAGS (Steyvers, 2018, version 1.3.2). We implemented a hierarchical Bayesian framework, in which the parameters for each subject are drawn from group-level gaussian distributions. We ran two chains with a burn-in period of 50,000 samples and thinning of two. We determined chain convergence of the chains such that $\hat{R} \leq 1.01$ [43]. For comparing the variants of the drift diffusion models, we ranked them according to the deviance information criterion [44, DIC].

**Posterior predictive response time distributions.** To ensure that the best-fitting model reflects and reproduces the observed data, we simulated 10,000 datasets based on the posterior distributions of the respective hierarchical model. For each individual participant, the model-predicted RT distributions were smoothed with a kernel smoothing function using density estimation (using MATLAB's `ksdensity` function) and overlaid onto the observed RT distributions.

## Results

### Model-free analyses

The participants made significantly more LL selections in the *high* (M = 63.00, SD = 19.28) as compared to the *low* (M = 53.30, SD = 21.09) magnitude condition (t(39) = -10.12, p < .001, one-tailed), reflecting the predicted magnitude effect. However, such a magnitude effect was not present in the response time patterns. The mean RTs were not significantly different between the *low* (M = 3.03, SD = 0.69) and *high* (M = 3.03, SD = 0.67) condition (t(39) = -0.01, p = .498, one-tailed).

### Softmax choice rule

We modelled the choices using the softmax choice rule, using both the absolute and normalised reward values. As hypothesised, we found a magnitude effect for temporal discounting, indicated by the negative shift parameter $s_{log(k)}$, which models the change in $log(k)$ from the *low* to the *high* condition (see Table 1). We observed a close correspondence of the parameter estimates from the softmax model based on absolute vs. normalised values, except for $\beta$ (see Fig 1), which scales with the value differences (see Eq 2).

### Drift diffusion modelling

**Model comparison.** We compared the fit of different variants of the DDM, including models with a linear (DDM_lin) and non-linear scaling (DDM_sig, and DDM_sig-shift) of the drift rate by the subjective value differences, and a model including parameters to model changes in the parameter distributions from the *low* to *high* condition (DDM_sig-shift). As a baseline

**Table 1. Group-level mean estimates and 95% HDIs of $log(k)$, $s_{log(k)}$ and $\beta$ using the softmax choice rule.**

|  | SM_abs | SM_norm |
|---|---|---|
| $log(k)$ | -4.44 (-5.07 to -3.79) | -4.44 (-5.04 to -3.82) |
| $s_{log(k)}$ | -0.80 (-0.90 to -0.70) | -0.74 (-0.86 to -0.64) |
| $\beta$ | 0.43 (0.05 to 0.72) | 27.23 (14.80 to 38.54) |

HDI: highest density interval; SM_abs: softmax model using absolute values; SM_norm: softmax model using normalised values; $log(k)$: discount rate (in logarithmic space); $s_{log(k)}$: shift parameter for the changes in $log(k)$ value from the *low* to *high* magnitude condition; $\beta$: inverse temperature parameter.

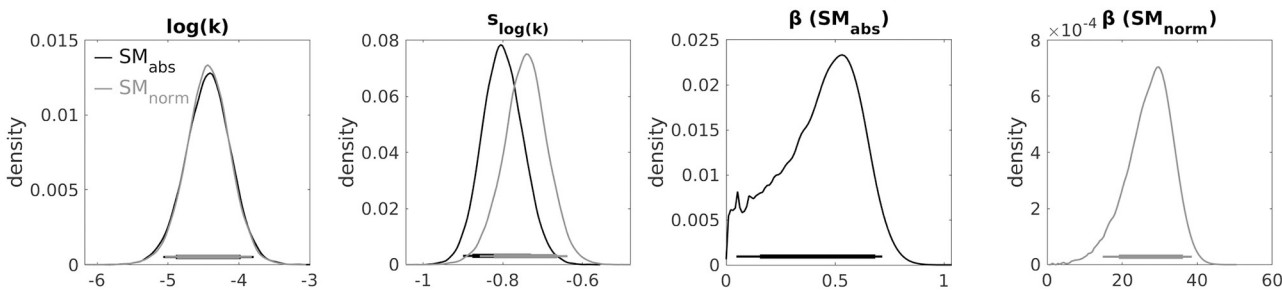

**Fig 1. Posterior distributions of the group-level parameter means from the softmax models based on absolute ($SM_{abs}$) and normalised ($SM_{norm}$) values.** $log(k)$: discounting parameter, $s$: shift in $log(k)$, $\beta$: inverse temperature parameter. Horizontal solid lines indicate the 85% and 95% highest density interval.

**Table 2. Model comparison of the variants of the drift diffusion models of temporal discounting using absolute values.**

|  | Value scaling | Value function | DIC | Rank |
|---|---|---|---|---|
| $DDM_0$ | - | - | 26830 | 4 |
| $DDM_{lin}$ | Linear | Hyperbolic | 24769 | 3 |
| $DDM_{sig}$ | Sigmoid | Hyperbolic | 22213 | 2 |
| $DDM_{sig-shift}$ | Sigmoid | Hyperbolic + shift | 22179 | 1 |

DIC = deviance information criterion; 0: no value scaling of the drift rate; lin: linear value scaling of the drift rate; sig: sigmoid value scaling of the drift rate. The $DDM_{sig-shift}$ includes additional shift parameters for $\alpha$, $\tau$, $z$, $v$, $v_{coeff}$, and $v_{max}$ to models changes from the *low* to *high* condition.

comparison, we formulated a model comprising no value modulation (constant drift rate, $DDM_0$). Further, we assessed the fit of all models using absolute vs. normalised values (see section Computational modelling of behaviour). The models implementing a non-linear scaling of the drift rate provided a superior fit to the data compared to models with a linear scaling. This was true for models operating on absolute *and* normalised values. Also, both the linear and non-linear models provided a superior fit compared to the $DDM_0$, see Tables 2 and 3.

Comparing the models based on absolute vs. normalised values, we observed a good correspondence of all model parameters, with the exception of $v_{coeff}$ and $s_{v_{coeff}}$, which of course scale directly with value differences.

**Posterior predictive response time distributions.** To verify that the best-fitting model can reproduce the observed RT distributions, we examined the posterior predictive RT distributions per participan. The posterior predictive RT distributions of the $DDM_{sig-shift}$ (using normalised values), along with the observed response time distributions, are depicted in Fig 2 (see Fig A in S1 Text of the supplementary material for the posterior predictive response time

**Table 3. Model comparison of the variants of the drift diffusion models of temporal discounting using normalised values.**

|  | Value scaling | Value function | DIC | Rank |
|---|---|---|---|---|
| $DDM_0$ | - | - | 26830 | 4 |
| $DDM_{lin}$ | Linear | Hyperbolic | 24286 | 3 |
| $DDM_{sig}$ | Sigmoid | Hyperbolic | 22210 | 2 |
| $DDM_{sig-shift}$ | Sigmoid | Hyperbolic + shift | 22170 | 1 |

DIC = deviance information criterion; 0: no value scaling of the drift rate; lin: linear value scaling of the drift rate; sig: sigmoid value scaling of the drift rate. The $DDM_{sig-shift}$ includes additional shift parameters for $\alpha$, $\tau$, $z$, $v$, $v_{coeff}$, and $v_{max}$ to models changes from the *low* to *high* condition.

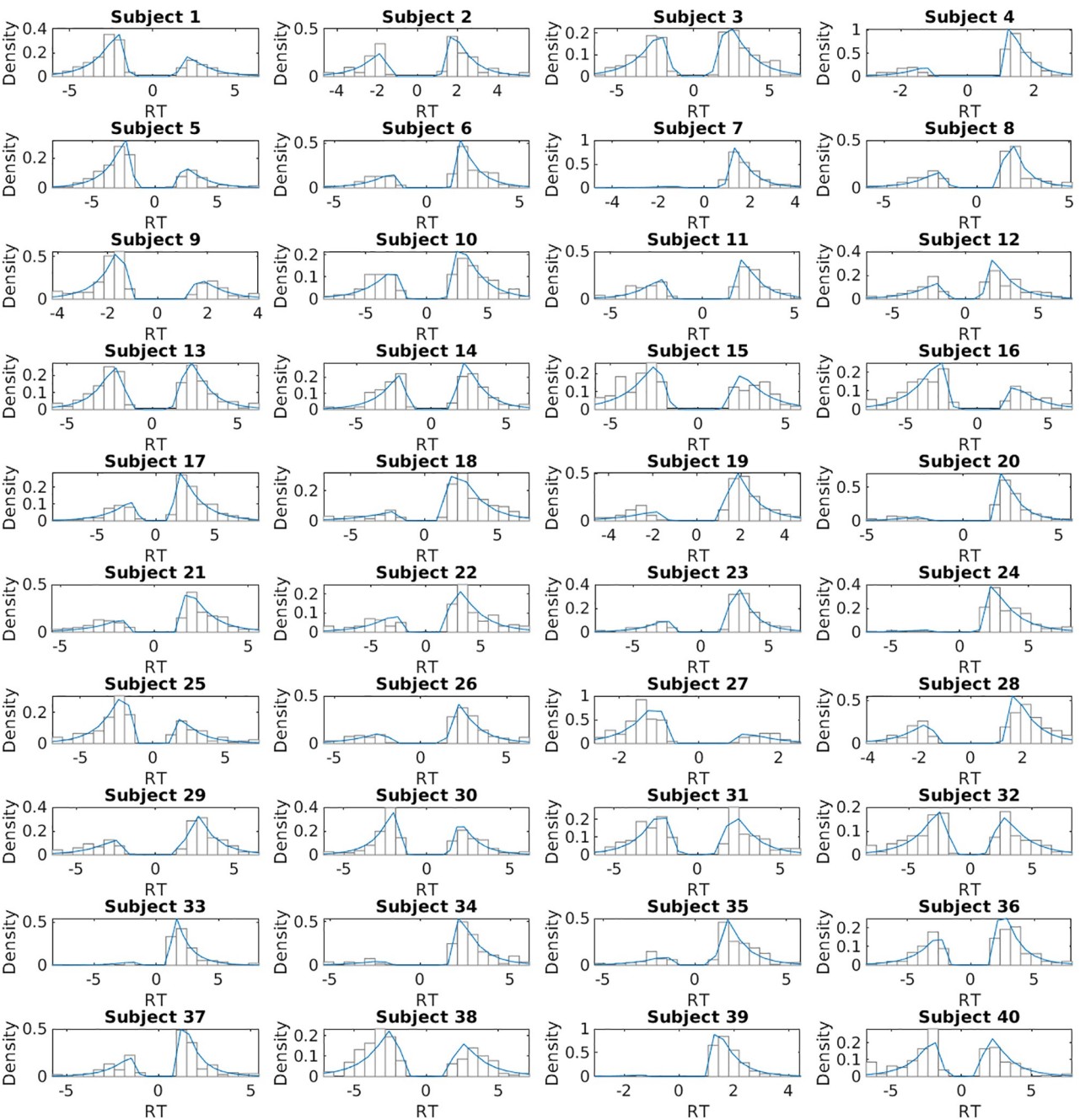

**Fig 2. Posterior predictive response time distributions (in blue) of the DDM$_{sig\text{-}shift}$ (using normalised values) for each participant, overlaid on the histograms of the observed RT distributions.** The negative response times arise from the boundary definitions of the DDM. We defined the lower boundary as choosing the SS reward, and the upper boundary as choosing the LL reward. For this purpose, choices towards the lower boundary were multiplied by -1. Negative response times indicate SS choices, positive response times indicate LL choices.

distributions of the DDM$_{sig\text{-}shift}$ using absolute values). The comparison showed that the model captures the characteristics of the response time distributions well.

**Analysis of model parameters.** We observed a positive association between the value differences and trial-wise drift rates, as indicated by the consistently positive drift rate coefficient parameter $v_{coeff}$ (see Tables 4 and 5).

**Table 4. Parameter group means and 95% HDIs of the posterior distributions of the drift diffusion models using absolute values.**

| | $DDM_0$ | $DDM_{lin}$ | $DDM_{sig}$ | $DDM_{sig\text{-}shift}$ |
|---|---|---|---|---|
| $\alpha$ | 2.72 (2.59 to 2.86) | 2.90 (2.76 to 3.04) | 3.27 (3.09 to 3.43) | 3.23 (3.05 to 3.41) |
| $s_\alpha$ | - | - | - | 0.09 (-0.01 to 0.19) |
| $\tau$ | 1.30 (1.22 to 1.39) | 1.30 (1.23 to 1.39) | 1.23 (1.15 to 1.31) | 1.23 (1.15 to 1.31) |
| $s_\tau$ | - | - | - | 0.01 (-0.01 to 0.04) |
| $z$ | 0.53 (0.52 to 0.55) | 0.53 (0.51 to 0.56) | 0.51 (0.49 to 0.52) | 0.50 (0.49 to 0.52) |
| $s_z$ | - | - | - | 0.02 (0.00 to 0.03) |
| $v$ | 0.18 (0.04 to 0.31) | - | - | - |
| $v_{coeff}$ | - | 0.05 (0.04 to 0.05) | 0.77 (0.61 to 0.94) | 0.78 (0.62 to 0.95) |
| $s_{v_{coeff}}$ | - | - | - | -0.08 (-0.17 to 0.01) |
| $v_{max}$ | - | - | 1.07 (0.98 to 1.16) | 1.10 (1.01 to 1.19) |
| $s_{v_{max}}$ | - | - | - | -0.04 (-0.11 to 0.03) |
| $log(k)$ | - | -4.42 (-5.08 to -3.78) | -4.45 (-5.07 to -3.83) | -4.47 (-5.07 to -3.85) |
| $s_{log(k)}$ | - | -0.52 (-0.71 to -0.34) | -0.82 (-0.93 to -0.71) | -0.77 (-0.89 to -0.65) |

HDI: highest density interval; 0: no value scaling of the drift rate; lin: linear value scaling of the drift rate; sig: sigmoid value scaling of the drift rate; $\alpha$: boundary separation; $\tau$: non-decision time; $z$: starting-point bias; $v$: drift rate; $v_{coeff}$: value difference to drift rate mapping; $v_{max}$: asymptote for $v$; $log(k)$: discount rate (in logarithmic space); $s$: shift parameter for the changes in parameter value from the *low* to *high* magnitude condition.

**Magnitude effects on model parameters**. For all models with value modulation of the drift rate, we observed an effect of reward magnitude on $s_{log(k)}$ (see Table 4), reflecting reduced discounting in the *high* compared to the *low* magnitude condition. This was also true for the models operating on normalised values (see Table 5). The starting point parameter $z$ was close to 0.5, indicating no strong bias towards either decision boundary, (SS rewards), with a rather small shift towards the upper boundary in the *high* magnitude condition. The effects of reward magnitude on the other parameters were negligible.

**Table 5. Parameter group means and 95% HDIs of the posterior distributions of the drift diffusion models using normalised values.**

| | $DDM_0$ | $DDM_{lin}$ | $DDM_{sig}$ | $DDM_{sig\text{-}shift}$ |
|---|---|---|---|---|
| $\alpha$ | 2.72 (2.59 to 2.86) | 2.95 (2.81 to 3.09) | 3.27 (3.10 to 3.45) | 3.24 (3.05 to 3.41) |
| $s_\alpha$ | - | - | - | 0.07 (-0.03 to 0.18) |
| $\tau$ | 1.30 (1.22 to 1.39) | 1.30 (1.22 to 1.38) | 1.23 (1.15 to 1.31) | 1.23 (1.15 to 1.31) |
| $s_\tau$ | - | - | - | 0.02 (-0.01 to 0.04) |
| $z$ | 0.53 (0.52 to 0.55) | 0.53 (0.51 to 0.56) | 0.51 (0.50 to 0.52) | 0.50 (0.49 to 0.52) |
| $s_z$ | - | - | - | 0.02 (0.00 to 0.03) |
| $v$ | 0.18 (0.04 to 0.31) | - | - | - |
| $v_{coeff}$ | - | 3.00 (2.68 to 3.32) | 39.44 (31.68 to 47.92) | 38.11 (30.11 to 46.90) |
| $s_{v_{coeff}}$ | - | - | - | 2.58 (-0.24 to 4.40) |
| $v_{max}$ | - | - | 1.07 (0.99 to 1.16) | 1.05 (0.96 to 1.14) |
| $s_{v_{max}}$ | - | - | - | 0.04 (-0.02 to 0.10) |
| $log(k)$ | - | -4.19 (-4.78 to -3.60) | -4.48 (-5.13 to -3.87) | -4.49 (-5.14 to -3.88) |
| $s_{log(k)}$ | - | -0.75 (-0.90 to -0.60) | -0.79 (-0.92 to -0.68) | -0.76 (-0.89 to -0.64) |

HDI: highest density interval; 0: no value scaling of the drift rate; lin: linear value scaling of the drift rate; sig: sigmoid value scaling of the drift rate; $\alpha$: boundary separation; $\tau$: non-decision time; $z$: starting-point bias; $v$: drift rate; $v_{coeff}$: value difference to drift rate mapping; $v_{max}$: asymptote for $v$; $log(k)$: discount rate (in logarithmic space); $s$: shift parameter for the changes in parameter value from the *low* to *high* magnitude condition.

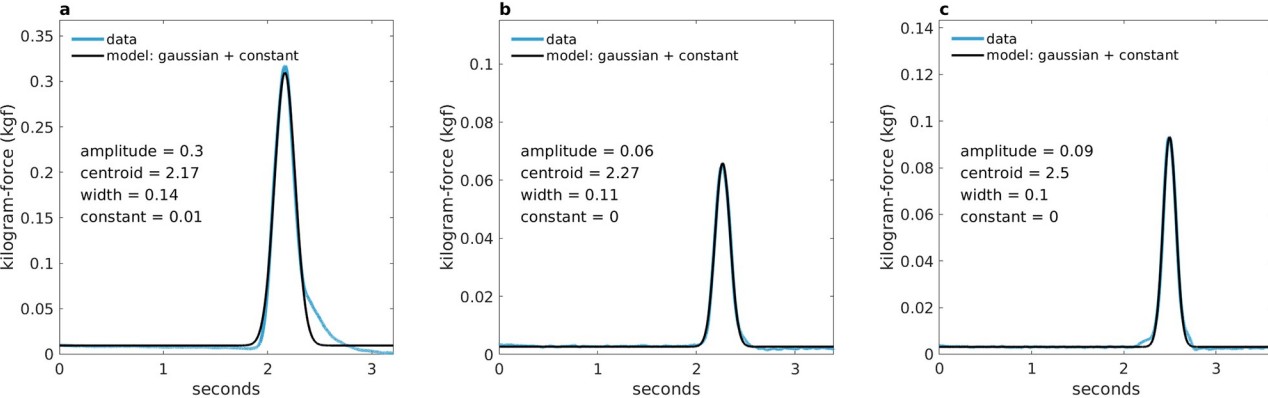

**Fig 3. Grip response and grip response model of three trials from a single participant.** The grip responses were modelled with a Gaussian function with parameters for amplitude, centroid, width, and a constant. A: trial 14, b: trial 152, c: trial 180. *R*-squared = 0.983, 0.998 and 0.997, respectively. Blue: preprocessed (baseline-corrected, normalised to maximal voluntary contraction [MVC], and smoothed) grip response data, black: modelled grip response. The unit of the x-axis and centroid parameter has been converted to seconds (sampling frequency: 2000 Hz, 1 s = 2000 samples).

**Effects of value normalisation**. Comparing the models based on absolute vs. normalised values, we observed a good correspondence of all model parameters, with the exception of $v_{coeff}$ and $s_{v_{coeff}}$, which of course scale directly with value differences.

## Motor response vigour (grip force)

The time between visual preselection of an option and choice registration with the grip force transducer was on average 0.03 seconds (*SD* = 0.11). The grip force responses were modelled with a Gaussian function (see Fig 3, 1 term plus constant, mean (range) goodness-of-fit across all trials and participants: *R*-squared = 0.98 (0.29–1.00), adjusted *R*-squared = 0.98 (0.29–1.00), root-mean-square error = 0.004 (0.0002–0.16). The parameter means (amplitude, centroid and width) per condition (*low*, *high*) are listed in Table 6, for mean values per participant and condition, and within-subject differences see Figs B and C in S1 Text of the supplementary material.

Since the data were non-normal (as assessed with Lilliefors tests yielding $p < .001$ for all tests), we performed Wilcoxon signed-rank tests to check for parameter differences between the *low* and *high* condition. In line with our preregistered hypothesis, the amplitude was significantly higher for the *high* compared to the *low* condition ($z = 1.90$, $p = .029$, one-tailed). In contrast to our preregistered hypothesis, the centroid, and also the width, did not differ between conditions ($z = 0.73$, $p = .768$, one-tailed, and $z = 1.75$, $p = .081$, two-tailed).

**Decision conflict effects.   Conflict based on choice probability (softmax model)**. Our first operationalisation of response conflict was based on the softmax choice probabilities.

**Table 6. Parameters of the gaussian-modelled grip force responses (means and standard deviations).**

|  | *Low* condition | *High* condition |
|---|---|---|
| **Amplitude** | 0.2142 (0.1414) | 0.2179 (0.1448) |
| **Centroid** | 3.15 (1.54) | 3.15 (1.52) |
| **Width** | 0.13 (0.05) | 0.13 (0.05) |

Amplitude has been normalised to MVC (maximal voluntary contraction). Centroid and width are reported in seconds.

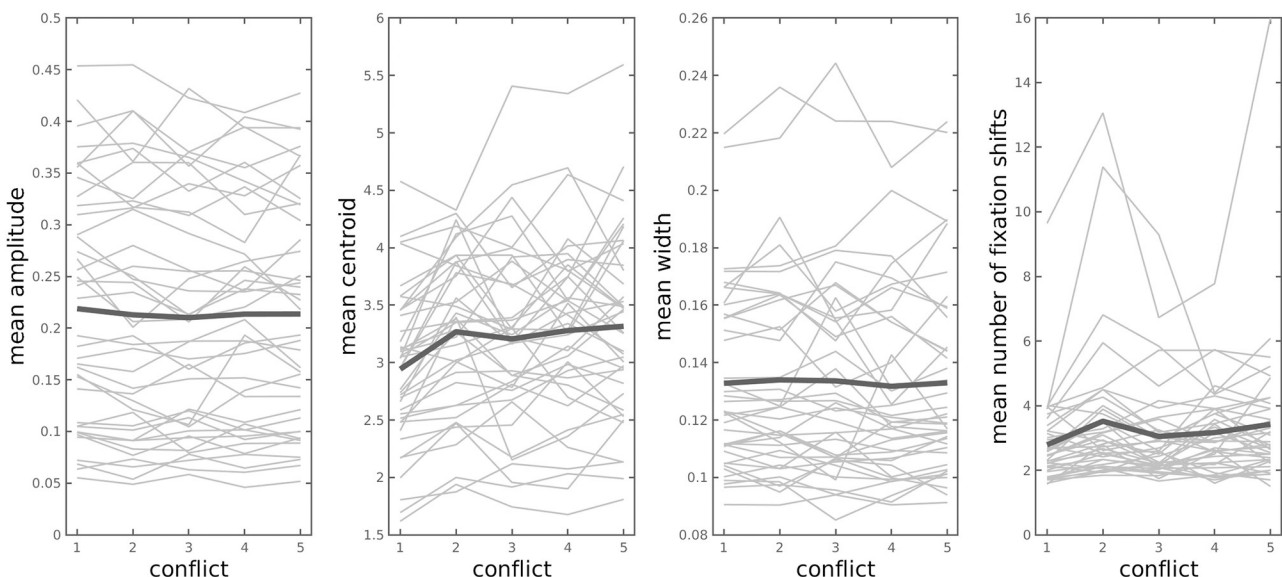

**Fig 4. Mean amplitude, centroid and width of the Gaussian-modelled grip force response and mean number of fixation shifts (from SS to LL, and vice versa) for trials of a given (binned) response conflict for each participant.** Thick lines depict the mean values across participants. Conflict is defined from 1 (low conflict) to 5 (high conflict), with a probability of .5 of choosing the LL reward as maximum conflict. Amplitude has been normalised to MVC (maximal voluntary contraction). Centroid and width are reported in seconds.

Because condition effects are more straightforward to interpret in the normalised model (see section Softmax choice rule), the following analyses are based on this model. The mean values for amplitude, centroid and number of fixation shifts for trials of a given response conflict (binned from 1 to 5) are depicted participant-wise in Fig 4 and listed in Table 7. The Bayesian regression is based on a continuous conflict measure (probabilities > 0.5 are 'flipped' to provide a common scaling from low to high conflict, whereby .5 represents the maximum conflict). The posterior distributions of the group-level parameter means for the regression coefficients are depicted in Fig 5 (medians: $\alpha$ = 0.12 [intercept] $\beta_1$ = -0.01 [amplitude], $\beta_2$ = 0.02 [centroid], $\beta_3$ = -0.004 [width], $\beta_4$ = 0.001 [$N$ fixation shifts]).

The Bayes factors for the regression coefficients for amplitude, centroid, and width of the grip response, and for the numbers of fixation shifts provide only anecdotal evidence for values greater than zero vs. smaller than zero (BF for $\beta_1$: 0.95, BF for $\beta_2$: 1.25, BF for $\beta_3$: 0.97, BF for $\beta_4$: 1.09). Since the 95% HDIs of all the posterior distributions fall neither completely inside nor outside the ROPE, we remain undecided for all three $\beta$ regression coefficients.

**Conflict based on subjective value differences (DDM).** The second operationalisation of response conflict was based on the trial-wise drift rate calculated based on the estimated

**Table 7. Mean amplitude, centroid, width and number of fixation shifts per conflict bin.**

|  | 1 | 2 | 3 | 4 | 5 |
|---|---|---|---|---|---|
| **Amplitude** | 0.2188 | 0.2128 | 0.2102 | 0.2135 | 0.2137 |
| **Centroid** | 2.9427 | 3.2680 | 3.2063 | 3.2789 | 3.3160 |
| **Width** | 0.1328 | 0.1340 | 0.1336 | 0.1317 | 0.1330 |
| **Fixation shifts** | 2.79 | 3.52 | 3.05 | 3.17 | 3.42 |

Conflict is defined from 1 (low conflict) to 5 (high conflict), with a probability of .5 of choosing the LL reward as maximum conflict. Amplitude has been normalised to MVC (maximal voluntary contraction). Centroid and width are reported in seconds.

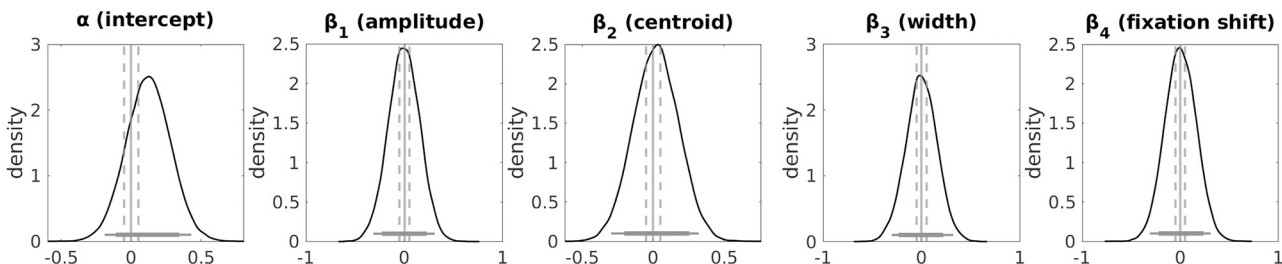

**Fig 5. Hierarchical Bayesian regression results. Regression of the parameters of the Gaussian-modelled grip force response and number of fixation shifts onto the trial-wise response conflict based on the choice probabilities (softmax model).** Posterior distributions of the group-level parameter means. $\alpha$: intercept, $\beta 1$: coefficient for amplitude, $\beta_2$: coefficient for centroid, $\beta_3$: coefficient for width, $\beta_4$: coefficient for fixation shift. Horizontal solid lines indicate the 85% and 95% highest density interval. Vertical solid lines indicate x = 0, and vertical dashed lines indicate the lower and upper bounds of the region of practical equivalence (ROPE).

parameters of the highest-ranked DDM using normalised values (DDM$_{sig-shift}$). Since we found no evidence that any of the regression coefficients for motor response vigour and number of fixation shifts were greater than vs. smaller than zero (or vice versa), we refer the reader to Section 3 and Fig D in S1 Text of the supplementary material.

Finally, we regressed the estimated grip force parameters amplitude, centroid and width, and the number of fixation shifts onto the subjective value differences between the (discounted) LL and SS rewards, based on the subject-specific *k* parameters of the highest-ranked model using absolute values (DDM$_{sig-shift}$). Recall that the analysis of the magnitude effect yielded an effect of condition, i.e. higher grip force amplitudes in the *high* compared to the *low* condition. Because condition differences in reward magnitudes are eliminated in the DDM based on normalised values (see Section 4 and Fig E in S1 Text of the supplementary material), the regression on subjective value differences is based on the DDM using absolute values.

The mean values for amplitude, centroid and number of fixation shifts for trials of a given value difference bin are depicted participant-wise in Fig 6. The posterior distributions of the group-level parameter means for the regression coefficients are depicted in Fig 7. The medians of the group-level posterior distributions were as follows: $\alpha$ = 3.33 (intercept) $\beta_1$ = 0.46 (amplitude), $\beta_2$ = -1.20 (centroid), $\beta_3$ = 0.19 (width), $\beta_4$ = -0.48 (*N* fixation shifts).

The Bayes factors provide very strong evidence that the coefficient for amplitude is greater than zero vs. smaller than zero (BF for $\beta_1$: 79.50), extreme evidence that the coefficient for centroid is below zero vs. above zero (BF for $\beta_2$: $> 10^{308}$), moderate evidence that the regression coefficient for grip force width is greater than zero vs. smaller than zero (BF for $\beta_3$: 5.06), and very strong evidence that the coefficient for number of fixation shifts is smaller vs. greater than zero (BF for $\beta_4$: 69.61). For $\beta_3$ we remain undecided, since the 95% HDI of the posterior distribution is neither completely inside nor outside the ROPE. For $\beta_2$ we reject the null value (95% HDI of posterior distribution entirely outside ROPE). For $\beta_1$ and $\beta_4$ we also reject the null value, since the 95% HDIs do not include zero and only 0.23% and 1.41%, respectively, of the 95% HDI overlap with the ROPE. This indicates higher grip force amplitudes, faster response times and a lower number of fixation shifts for trials with higher subjective value differences between the options.

**Value sum effects.** To analyse the association between motor response vigour, fixation shifts and total value, we regressed the parameters of the Gaussian grip force model and the number of fixation shifts onto the sum of the subjective LL and SS option values (based on the drift diffusion model using absolute subjective values). The medians of the group-level posterior distributions were as follows: $\alpha$ = 0.59 (intercept), $\beta_1$ = 0.62 (amplitude), $\beta_2$ = -0.47

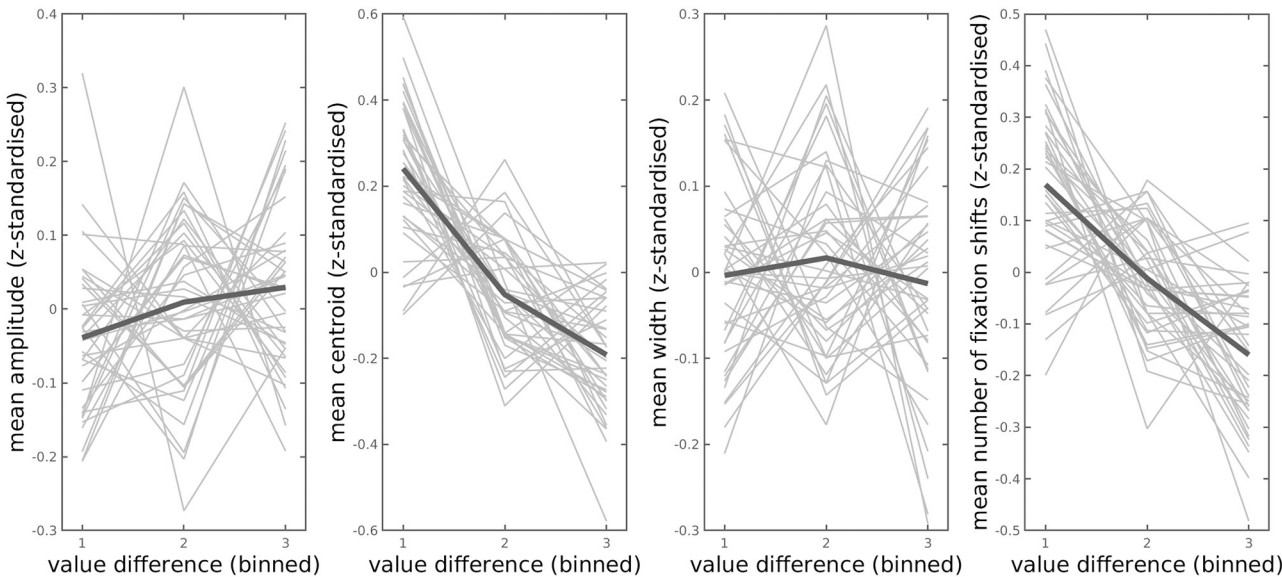

**Fig 6. Mean amplitude and centroid of the Gaussian-modelled grip force response and mean number of fixation shifts (from SS to LL, and vice versa) for trials of a given value difference bin.** The (absolute) value differences were *z*-standardised and binned participant-wise into 3 groups of equal size (based on quantile ranks of the values, 1: lower value differences, 3: higher value differences). Thick lines depict the mean values across participants.

(centroid), $\beta_3 = 0.01$ (width), $\beta_4 = -1.11$ (*N* fixation shifts) (see Fig 8). The Bayes factor for the regression coefficient for amplitude provides extreme evidence for values greater than zero vs. smaller than zero (BF for $\beta_1$: 111.18). For the centroid coefficient, the Bayes factor provides strong evidence for values smaller than zero vs. larger than zero (BF for $\beta_2$: 16.37). For the width coefficient, the Bayes factor provides only anecdotal evidence for values greater than zero vs. smaller than zero (BF for $\beta_3$: 1.04). The regression coefficient for fixation shifts provides extreme evidence for values smaller than zero vs. larger than zero (BF for $\beta_4$: 10933.53).

For $\beta 1$ and $\beta_4$ we reject the null value (95% HDI of posterior distribution entirely outside ROPE). For $\beta_2$ and $\beta_3$ we remain undecided, since the 95% HDI of the posterior distribution is neither completely inside nor outside the ROPE. Accordingly, this analysis shows higher grip force amplitudes and fewer fixation shifts for trials with high value sums across the options. Running separate regressions on value sums of the *low* and *high* magnitude condition revealed

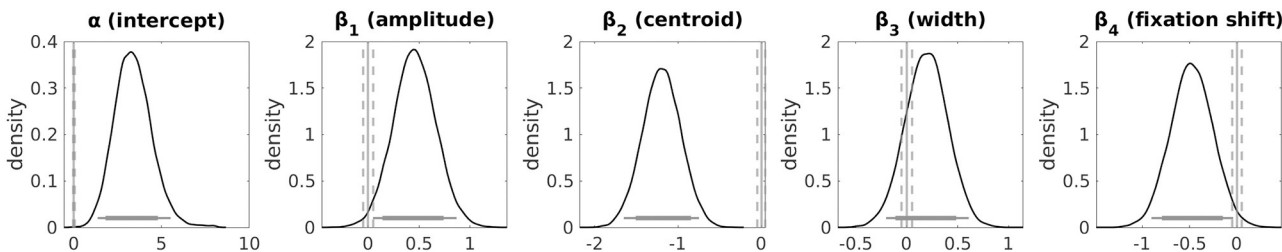

**Fig 7. Hierarchical Bayesian regression results. Regression of the parameters of the Gaussian-modelled grip force response and number of fixation shifts onto the subjective value differences (DDM).** Posterior distributions of the group-level parameter means. $\alpha$: intercept, $\beta 1$: coefficient for grip force amplitude, $\beta_2$: coefficient for grip force centroid, $\beta_3$: coefficient for grip force width, $\beta_4$: coefficient for fixation shift. Horizontal solid lines indicate the 85% and 95% highest density interval. Vertical solid lines indicate x = 0, and vertical dashed lines indicate the lower and upper bounds of the region of practical equivalence (ROPE).

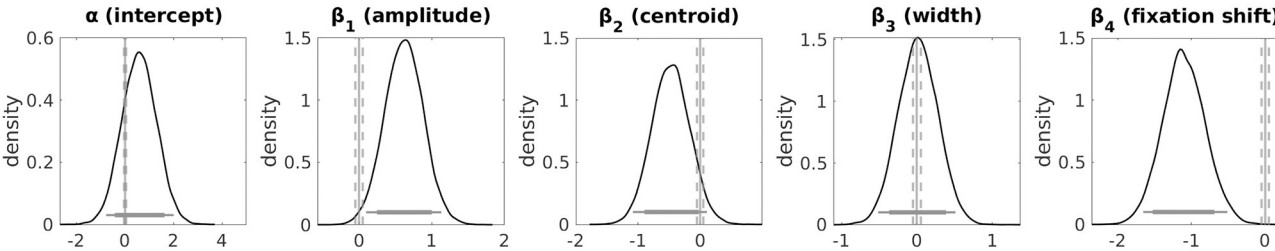

**Fig 8. Hierarchical Bayesian regression results. Regression of the parameters of the Gaussian-modelled grip force response and number of fixation shifts onto the total sum of the subjective option values from the drift diffusion model (DDM).** Posterior distributions of the group-level parameter means. $\alpha$: intercept, $\beta1$: coefficient for grip force amplitude, $\beta_2$: coefficient for grip force centroid, $\beta_3$: coefficient for grip force width, $\beta_4$: coefficient for fixation shift. Horizontal solid lines indicate the 85% and 95% highest density interval. Vertical solid lines indicate x = 0, and vertical dashed lines indicate the lower and upper bounds of the region of practical equivalence (ROPE).

that this effect was driven by the *high* magnitude condition (see Section 6, Figs G and H in S1 Text of the supplementary material).

## Discussion

We explored whether value computation and response conflicts during intertemporal choice are reflected in motor response vigour and visual fixation patterns. For this purpose, we measured the handgrip force applied during choice, and the concurrent fixation shift patterns between the choice options. Assuming hyperbolic discounting, we compared variants of the softmax and drift diffusion model and assessed the relationship between the estimated model parameters, motor response vigour and fixation shifts. The intertemporal choice task comprised two conditions, a *low* and *high* magnitude condition (*low* vs. *high* SS reward), which allowed us to directly assess the impact of overall smaller vs. larger reward magnitudes on response vigour. To represent both conditions in a single model, we included shift parameters to model the changes in parameter values from the *low* to the *high* magnitude condition.

We compared models with a linear and non-linear (sigmoid) modulation of the drift rate by the subjective value differences, and, since the drift rate parameter is dependent on the absolute magnitude of the options' values, models using absolute vs. normalised option values. We then analysed the relationship between decision conflict and response vigour, in particular the trial-wise amplitude, centroid and width of the Gaussian-modelled grip force response and the number of fixation shifts between the options. Further, we investigated if the magnitude effect, which describes reduced discounting for higher amounts [22, 23], is also reflected in the grip force strength.

As hypothesised, participants discounted rewards as a function of delay. The choice and response time (RT) data were best accounted for by a DDM including a non-linear modulation of the drift rate by the subjective value differences. As in previous studies, [23, 29], and in accordance with our hypothesis, we found a magnitude effect for temporal discounting, indicating that higher rewards were discounted less. This effect was also evident in motor response vigour: higher forces were applied in the *high* vs. the *low* magnitude condition. In addition, trials with higher subjective value differences between the options were associated with higher grip forces, faster response times and a lower number of fixation shifts.

In general, the estimated non-decision times $\tau$ were longer than in typical laboratory experimental setups ($>$ 1000 ms) [19, 29]. The non-decision time parameter $\tau$ models time that is not related to the decision process, such as stimulus encoding and motor response execution. Our estimated non-decision times were comparable to the estimated non-decision times from

a recent study using a VR environment, where the participants logged their responses using VR-compatible controllers, as opposed to simple response keys [45]. This is likely due to the task's requirement of first preselecting an option through visual fixation before finally selecting it using the hand dynamometer. However, since $\tau$ reflects both motor and non-motor components, which of these processes is affected cannot be inferred from $\tau$ alone. There is preliminary work on the decomposition of the non-decision time of drift diffusion models using electro-myographical activity [46]. The authors conclude that stimulus encoding does not necessarily end when evidence accumulation begins, and that the onset of the motor response does not necessarily denote the end of the deliberation process. Further, the non-decision time may be influenced by participants' adjustments in response to task instructions. Therefore, a meaningful comparison of non-decision times across experiments with different response schemes may only be made if all other experimental parameters are kept constant. Still, applying the DDM works well in settings with different task demands and response modes.

## Model comparison

The choice and RT data were best accounted for by a drift diffusion model specifying a non-linear mapping between the subjective value differences and trial-wise drift rates. Following the DIC criterion, the variants of the DDMs implementing a transformation of the scaled value differences using a sigmoid function [26] provided a superior fit to the data compared to both the DDM using a linear modulation and the DDM involving no value modulation. We found a close correspondence between the observed response time distributions and the response time distributions simulated using the estimated posterior parameter distributions, demonstrating that the best-fitting model captured the characteristics of the response time distributions reasonably well.

## Magnitude effect

Replicating previous findings [22, 23, 29], and in accordance with our hypothesis, we found a magnitude effect for temporal discounting, such that higher rewards were discounted less. While the model-free analysis revealed more LL choices in the *high* compared to the *low* magnitude condition, the magnitude effect was further reflected in the $log(k)_{shift}$ parameter, which was consistently negative in all variants of the softmax and drift diffusion models. Importantly, as predicted, this magnitude effect was also reflected in motor response vigour: Looking at the amplitude parameter of the Gaussian-modelled grip force response, we found that stronger forces were exerted in the *high* compared to the *low* magnitude condition. Contrary to our hypothesis, the RTs were not significantly different between the two conditions. The effect of reward magnitude on the discount rate (reduced discounting for higher rewards) appears to be a consistent effect [22, 23, 29], and our data reveal that this effect is reflected in both choice behaviour and motor response vigour (grip force amplitude) during response selection.

Based on this finding, we carried out a further exploratory analysis, replacing subjective value differences with total value. In line with the idea that subjective (rather than objective) option dimensions shape behaviour [7], the associations with amplitude and centroid were more pronounced for the model that used subjective (DDM-based) rather than objective (absolute magnitude) values for the computation of total value (see section Value sum effects and Section 5 and Fig F in S1 Text of the supplementary material). Further, comparing the results for conflict and subjective value differences and total value, respectively, it appears that total value was most strongly associated with grip force amplitude and number of fixation shifts. These effects share some similarity with other modulatory effects of pavlovian cues, such as pavlovian instrumental transfer, where conditioned stimuli affect the vigour with

which an action is executed [47, 48]. However, the present effects of value sum on motor response vigour were not instrumental, as grip force was decoupled from outcome.

### Decision conflict effects

First, we carried out a model-based analysis of the trial-wise grip force time courses. A gaussian model, decomposing grip force time courses into amplitude, centroid and width parameters for each trial provided an excellent fit to the single-trial grip force trajectories (mean $R$-squared = .98). We then analysed the relationship between decision conflict, the grip force parameters and fixation shifts, operationalising decision conflict based on the choice probability as derived from the softmax choice rule, and based on the trial-wise drift rate, as derived from the best-fitting DDM. Contrary to our hypothesis, however, we found no relationship between decision conflict and motor response vigour, response times, or fixation shifts. However, regressing motor response vigour and fixation shifts directly on the subjective value differences (based on the estimated parameters of the best-fitting DDM, we found that the amplitude and centroid of the grip response, as well as the number of fixation shifts were significantly related to these. As predicted, grip force amplitudes increased, and response times (centroids) decreased with increasing subjective value differences between options. In addition, the number of fixation shifts decreased with increasing subjective value differences. Looking at the models regressing motor response vigour and fixation shifts onto subjective value differences and value sum, respectively, the negative relationship with the centroid parameter appeared to be most pronounced for value difference. In the model with objective value sum (model-free, see Section 5 and Fig F in S1 Text of the supplementary material) the effect was not visible at all. Hence, the centroid (response time) effect appeared to be relatively specific for response conflict, whereas the grip force amplitude effect was observed in both models, albeit more pronounced for the value sum model.

The null effects for the conflict measure based on the softmax model likely arise because for large value differences, the conflict predictor approaches zero. The second regression was based on the drift diffusion model using a non-linear (sigmoid) scaling of the drift rate by the subjective value differences, so we speculate that the null effects for conflict based on the drift rate arise because the drift rate does not scale linearly with the value differences (as the value difference exceeds an individual threshold, the corresponding drift rate is mapped to $v_{max}$). We therefore assume that the effects we found when regressing motor response vigour and fixation shifts directly on the subjective value differences are driven by trials with large absolute value differences. Taken together, these results suggest that the observed associations between value differences, grip force parameters and fixation patterns are driven by absolute value differences, rather than decision conflict.

This suggests that valuation or implicit motivation could be reflected in these measures. In contrast to Pessiglione and colleagues [15], where the force produced was related to the payout (reward height magnitude was presented subliminally), we kept the force produced unrelated to the payout. Therefore, even when the force produced is unrelated to the payout (and the participants are unaware that force production is being measured), it is nonetheless related to the subjective value difference and even more so, the value sum. [49, 50] In the present study, the participants applied more force in trials with higher value differences, and in particular a higher subjective value sum of the options. This suggests that motivational processes are also reflected in motor response vigour.

Summarising the findings with respect to our hypotheses, as expected, participants discounted rewards as a function of delay (hypothesis i). In accordance with our predictions, we further found evidence that differences in subjective utility modulated response times and grip

forces, such that response times decreased and grip forces increased with increasing subjective value differences between options (hypothesis ii). Unlike what we thought, we found no relationship between decision conflict and grip force, response time or fixation shifts (hypothesis iii). Yet, in line with our hypothesis, higher rewards were discounted less and elicited stronger effort (magnitude effect). Contrary to our hypothesis, however, the response times were not significantly different between the two magnitude conditions (hypothesis iv).

### Dopamine and response vigour

Although dopamine neurotransmission was not measured in the present study, the observed effects might be mediated by dopamine. A number of studies suggest that the anterior cingulate cortex and its dopaminergic pathways are involved in the integration of effort and reward [51–54]. Pharmacological enhancement of dopamine transmission increases the willingness of animals to accept delays and to expend effort to obtain rewards (for a review, see [55]). Three studies with human subjects also reported higher force production in states with augmented dopamine transmission [12–14]. In a rewarded odd-ball discrimination task, Beierholm and colleagues [56] demonstrated that L-DOPA modulated reward-related response vigour (reaction times). The results suggest that the influence of reward rate on response vigour is mediated by dopamine transmission. Further, augmented dopamine transmission increased response vigour (reduced reaction times) in a temporal discounting and reinforcement learning task [29, 57, 58]. In addition to dopamine, noradrenaline is also involved in force production [59] and conflict resolution [60, 61]. However, manipulating noradrenaline levels does not appear to affect reward sensitivity [62].

### Relevance

Our results suggest that in addition to choices and response times, measures of response vigour may provide information regarding valuation during intertemporal choice. Other tasks involving subjective evaluation of options may also conceivable. Since several maladaptive behaviours and psychiatric conditions, including impulsivity, substance use disorders and behavioural addictions, have been linked to increased discount rates (see, e.g. [4, 5, 63, 64]), this task is particularly interesting from a clinical perspective.

Using response vigour as an implicit measure of utility may open up the possibility to assess utility in cases where explicit reports are not possible, i.e. in different patient groups. In the area of statistical learning, patients with hippocampal damage show impairments in certain processes when the patients are required to explicitly report regularities or patterns [65, 66]. When testing for implicit knowledge in a motion discrimination task, patients with hippocampus damage showed a similar performance to controls [67]. Experimental approaches such as those employed in the present study might be informative in such patient populations.

### Limitations

Finally, there are some limitations to our study. For the present task, it would have been interesting to also include pupillometry and more comprehensive analyses of e.g. saccade reaction times and velocities [17, 68]. Since the usage of a force transducer functions as a single key, some method of preselecting one of two options was necessary. Choice selection was thus implemented such that an option was *pre*selected by visual fixation and selected by subsequently pressing the force transducer. Hence, an option could only be chosen if it was concurrently fixated, which may have restricted the fixation patterns. Therefore, we limited the analyses to the shifts of fixation. Further, since we did not specifically construct isoluminant

stimuli, the analysis of pupil dilation would be confounded by differences in luminance between the stimuli and conditions.

Although, based on the literature, an involvement of dopamine in the effects examined here is likely, dopamine neurotransmission was neither measured nor manipulated. Future studies would benefit from examining this in greater detail.

## Conclusion

In the present work, we investigated motor response vigour, specifically grip force applied during response selection, and fixation patterns as an implicit measures of subjective utility during intertemporal choice. Comparing variants of the drift diffusion model, we found that the choices and response times were best accounted for by a drift diffusion model that included a non-linear scaling of the drift rate by the subjective value differences. A magnitude effect for temporal discounting was apparent in both choice and motor response vigour, such that higher rewards were discounted less and selected with higher grip force. The magnitude effect was evident not only between conditions, but also in the form of an association between total value (sum of discounted values across conditions) and response vigour. Further, the peak forces (grip force amplitudes), response times (grip force centroids) and the number of fixation shifts were related to the subjective value differences between the options. Normalising the options' values across conditions eliminated these effects. We conclude that the effects were likely driven by large absolute (discounted) value differences between the options. A further exploratory analysis revealed that the subjective value sum across options showed an even more pronounced association with the trial-wise grip force amplitudes and number of fixation shifts. The force applied was unrelated to the payout and the participants were not informed that force production was measured. Nonetheless, it was related to the subjective value differences between the options, suggesting that valuation or implicit motivation is reflected in motor response vigour. Future studies might explore the extent to which neuropsychiatric disorders associated with impairments in decision-making and effort are likewise associated with changes in such implicit measures of motivation.

## Supporting information

**S1 Text.** Fig A: Posterior predictive response time distributions of the DDM$_{\text{sig-shift}}$ (using absolute values) for each participant, overlaid on the histograms of the observed RT distributions. Fig B: Parameters of the modelled grip response (mean values per participant and magnitude condition). Fig C: Within-subject differences of the parameters of the modelled grip response between the *low* and *high* magnitude condition. Fig D: Hierarchical Bayesian regression of the parameters of the Gaussian-modelled grip force response and number of fixation shifts onto the trial-wise response conflict based on the drift rate (DDM). Fig E: Hierarchical Bayesian regression of the parameters of the Gaussian-modelled grip force response and number of fixation shifts onto the subjective value differences (DDM). Fig F: Hierarchical Bayesian regression of the parameters of the Gaussian-modelled grip force response and number of fixation shifts onto the total sum of the option amounts (model-free). Fig G: Hierarchical Bayesian regression of the parameters of the Gaussian-modelled grip force response and number of fixation shifts onto the total sum of the subjective option values from the drift diffusion model (DDM) for the *low* magnitude condition. Fig H: Hierarchical Bayesian regression of the parameters of the Gaussian-modelled grip force response and number of fixation shifts onto the total sum of the subjective option values from the drift diffusion model (DDM) for the *high* magnitude condition. Section 3: Conflict based on trial-wise drift rate (DDM). Section 4: Conflict based on subjective value differences (DDM using normalised values). Section 5: Sum of

larger-later (LL) and smaller-sooner (SS) amounts (model-free). Section 6: Sum of the subjective larger-later (LL) and smaller-sooner (SS) option values (DDM) per magnitude condition. (PDF)

## Author Contributions

**Conceptualization:** Elke Smith, Jan Peters.

**Data curation:** Elke Smith.

**Formal analysis:** Elke Smith.

**Funding acquisition:** Jan Peters.

**Investigation:** Elke Smith.

**Supervision:** Jan Peters.

**Visualization:** Elke Smith.

**Writing – original draft:** Elke Smith.

**Writing – review & editing:** Elke Smith, Jan Peters.

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
