## [Decision Letter · Decision Letter 0]

14 Dec 2021

Dear Dr. Smith,

Thank you very much for submitting your manuscript "Motor response vigour and fixations reflect subjective preferences during intertemporal choice" for consideration at PLOS Computational Biology.

As with all papers reviewed by the journal, your manuscript was reviewed by members of the editorial board and by several independent reviewers. In light of the reviews (below this email), we would like to invite the resubmission of a significantly-revised version that takes into account the reviewers' comments.

In particular, the authors should pay careful attention to strengthening the analyses that establish the links between visual fixations / grip force, and the two computational variables, as the manuscripts claims hinge on this link. (Note: one of the reviewers' attachment was incomplete; we have attached the complete version IN ADDITION to the incomplete one)

We cannot make any decision about publication until we have seen the revised manuscript and your response to the reviewers' comments. Your revised manuscript is also likely to be sent to reviewers for further evaluation.

Sincerely,

Gunnar Blohm, Ph.D.

Associate Editor

PLOS Computational Biology

Samuel Gershman

Deputy Editor

PLOS Computational Biology

Reviewer's Responses to Questions

**Comments to the Authors:**

Reviewer #1: Uploaded as as attachment

Reviewer #2: The manuscript reports the results of a pre-registered study that investigated how implicit behavioral measures (gaze fixations and grip force) are affected by the subjective values of options presented during an intertemporal choice task. Of interest, there were links with the overall value of the option pair and the difference between option values.

Among the strengths of the paper is an original choice task in which options are selected by visual fixation and confirmed by pressing a handgrip. Another is the sophistication of the computational analyses and the use of Bayesian techniques for model selection. My global impression however is that the paper lacks focus and fails to identify the main questions of interest. As stated in my summary above, the analyses should target the links between the two implicit measures (visual fixations and grip force) and the two computational variables (value sum and difference). This could be done in a model-free analysis that would be straight and simple. In the current paper, these analyses are buried within tons of computational variants that address no meaningful issue at the conceptual level. For instance, whether the DDM provides a better account of RT when taking as input an affine or a sigmoid transformation of value difference will not change our understanding of how the brain works. My main suggestion would be to streamline the analyses around the questions that might bring some conceptual advance. I understand there is the issue of pre-registration, which constrains the authors to sticking on their first ideas. While I understand the virtue of pre-registration in the context of a replication study (where one wants protection against false positives), I think it is problematic in the context of an exploration study (where one wants protection against false negatives) that examines novel potential effects, as we have here. I would therefore encourage the authors to go beyond their pre-registered analyses (which could be moved in supplementary material) and explore their data in a more systematic approach.

More specific concerns:

1) The introduction does not provide a justification for the design. If the idea is to test the link between response vigor and subjective value, then why using intertemporal choice, and why manipulating reward so as to induce a magnitude effect? It also introduces useless or misleading concepts, such as effort discounting, that are never used later in the paper.

2) Among the four predictions, only the second and third ones (ii and iii) relate to the main question (correlates of value sum and difference). The first prediction, as it is phrased, is trivially false (participants do not always prefer smaller-sooner over larger-later rewards, it obviously depends on rewards and delays). The fourth prediction about the magnitude effect could be reduced to the second (whether value sum affects response vigor). The questions about whether delay and reward magnitude have an influence on choices, which are conflated with predictions (i) and (iv) could be side-tracked as already answered in many previous studies.

3) There are fluctuations in the notion of response vigor, which designates grip force specifically in some instances (such as in the abstract and title) and all non-instrumental measures (including number of fixations) in other instances (such as in the first sentence of the conclusion).

4) I have nothing against using DDM to predict RT, but in the context of this paper it seems like an unnecessary complication. The proxy for confidence (or inverse conflict), i.e. value difference, could be taken directly from the softmax function. Similarly, fitting a response function with 3 parameters to force pulses seems like a good idea, but probably weakens the analyses testing the link between grip force and value-related variables. This is because these correlated parameters compete to explain the same variable in the multiple regression model. I suggest using a model-free measure (like the area under the curve) for a more sensitive and straightforward analysis.

5) The discussion is essentially a reiteration of the results, while it should provide perspectives for the conceptual advances and limitations for the interpretations.

Minor concerns:

1) There are a lot of typos and mistakes throughout the manuscript, which should be checked carefully. For examples:

- Maximum conflict (page 9) should be 0.5 and not .05.

- Right graphs in Fig. 1 (page 12) should be labeled SMnorm and SNabs (and/or colored in grey and black)

- ‘shift’ should be deleted in the depiction of the DDMsig model in Table 2 (page 13)

- There are three parameters in the force response function but only two are reported in Table 7 and reported in Fig. 4 (page 19).

- Two studies are announced in the conclusion paragraph but three references are provided.

2) I would suggest to replace ‘fixations’ by ‘visual fixation pattern’ in the title, which would stand symmetrically to ‘motor response vigor’. Also, these measures did not reflect ‘subjective preferences’, as suggested in the title, but value sum and difference (unsigned). This is critical because some passages seem to suggest that one can know which option is preferred by looking at grip force, while this is clearly not the case.

3) Doubling the number of participants required by the power calculation means that the power calculation was not taken seriously. I agree that for an explorative study, testing a new effect, power calculation is meaningless (because one cannot know the expected effect size). But then providing a power calculation sounds like showing off superficial rigor.

4) The link with dopamine suggested in the discussion is interesting but not the only possibility. For instance, there is a literature relating conflict to noradrenalin (and anterior cingulate cortex) that the authors may want to cite.

5) The application to neuropsychiatric conditions is too far-fetched. Removing it would not undermine the interest of the study, in my opinion. Besides, that effort processing is impaired in Parkinson’s disease makes the grip force measure more problematic, not more relevant.

6) Some details are missing now and then.

- what is the unit for delays? (Presumably, days for the first and months for the last one)?

- What is the force threshold that must be reached to confirm a choice?

- How was conflict binned into 5 levels?

Reviewer #3: In this paper Smith and Peters use an inter temporal choice task to examine how non-voluntary (or non-instrumental) vigor of a motor response can be influenced by subjective estimation of reward options. Several previous studies have shown links between reward processing and vigor of motor responses, with a possible mediator of Dopamine.

Vigor was here measured through grip-force, while eye tracking was used to examine visual fixation.

The two main findings were

1. There is a magnitude effect causing higher rewards to be discounted less, and with higher vigor

2. For easier trials (large subjective value differences between options) had larger vigor, shorter response times, and few shifts in fixation between the options.

Neither of these results are very surprising but the paper is well written, and the methods all seem very solid. The results are interesting, and worth publishing in PLoS Comp Bio.

It would have been interesting if more fine grained results could have been squeezed out of the eye tracking data, but that may be for future work (latency in saccade initiation? pupilometry?).

I don’t have any major comments, but just wanted to raise on potential issue with the model specification.

Twelve parameters (for the largest model) is a lot, making me a bit worried about over-fitting. But as DIC is meant to correct for that, I will take the numbers at face value.

I have some minor comments:

Page 6

“ sixteen percentages of the SS reward value “ are these percentages or ratios?

Are the delay periods really [1 7 13 31 58 12], i.e. 12 as the last? Are these given in seconds? Hours? Days?

“track the gaze patterns and to give real-time feedback to the participant,” what feedback was given? Or was it just to highlight the current fixated reward option

“All logfiles were checked for stereotypic response patterns” should that be non-stereotypic? Maybe rephrase

Incidentally, having a figure showing the experimental setup (even if very simple) could be useful

Page 9

I did not understand how the decision conflict was operationalised as a number from 1 to 5. Please explain

Page 11

The variable d is used for both the grip model and the number of fixation shifts, please change one of them.

Page 12

Figure 1, make clearer that the two right most plots are for the same parameter but different models.

Page 15

Figure 2, I would suggest to make this bigger

Also, are the RTs split by L/R? I.e. why are there negative RTs?

Page 18

Table 6 and Fig 3, why report the centroid in number of samples instead of time?

Also, is there a reason gap force width is not reported here?

For the discussion on Dopamine or future work: The authors may also find Beierholm et al 2013 (https://www.nature.com/articles/npp201348) interesting, showing that Dopamine modulated reward-related vigour in a reaction time task, thus building on the work by Niv et al 2007 (https://link.springer.com/article/10.1007/s00213-006-0502-4).

A few of the references are incomplete, e.g. Moreira & Barbosa, as well as Schultz

Reviewer #4: PCOMBIOL-D-21-02032

This study investigates whether response vigor serves as an additional implicit measure of subjective utility during intertemporal choice. They find that motor response vigor reflects the difference in subjective value between the options and is also includence by the absolute value of the options. Subject grasp earlier and with more force the greater the difference int subjective value between the options and to some extent, the greater the absolute value of the two options.

This is an interesting and novel study. It has been previously shown that relative saccade vigor during deliberation between two options correlates with the difference in subjective value between the options. This study shows that grasp vigor when selecting between two options also correlates with the difference in subjective value. This demonstrates that vigor of multiple effectors (eyes, hand) may simultaneously reflect subjective utility. Overall, I am excited to see these results. However, I have a few comments and suggestions that I believe would strengthen and extend their findings.

General Comments

1. I am not sure why the authors do not simply measure peak grasp force as their main vigor measure. Instead there is a model that is fit to the profiles. I understand that they show there is a good fit between model and data, but It would be helpful if they also performed their same analysis using peak grasp force.

a. The authors could also consider looking at rate of grasp force development and time to required force as additional measures of vigor.

2. I think the magnitude effect on response vigor is quite interesting. As I understand it the magnitude effect suggests that for the two pairs of options with the same relative difference in subjective value, the pair with the greater magnitude will have a greater response vigor. Related to this, I have the following questions:

a. The authors seems to suggest that the magnitude effect on response vigor is in addition to the effect of the difference in subjective value, which is potentially very interesting. However, could the increase in vigor in the high magnitude condition be due to greater relative values differences between options in high magnitude condition compared to the low magnitude condition?

b. If indeed there is a magnitude effect then this could potentially be seen across all trials and not just between conditions. Do the authors see an effect of the total value (sum of both options) on response vigor?

Minor Comments

1. Please provide additional detail about the deliberation phase of the experiment. For example, how much time did the subjects have to indicate their choice?, how much time was allowed to pass between pre-selection and the grasp response?

**Have the authors made all data and (if applicable) computational code underlying the findings in their manuscript fully available?**

Reviewer #1: Yes

Reviewer #2: Yes

Reviewer #3: Yes

Reviewer #4: **No: **The subjects did not consent this, thus the data is available on a private server.

PLOS authors have the option to publish the peer review history of their article (what does this mean?). If published, this will include your full peer review and any attached files.

Reviewer #1: No

Reviewer #2: No

Reviewer #3: **Yes: **Ulrik R Beierholm

Reviewer #4: No
---

## [Decision Letter · Decision Letter 1]

8 Mar 2022

Dear Dr. Smith,

Thank you very much for submitting your manuscript "Motor response vigour and visual fixation patterns reflect subjective valuation during intertemporal choice" for consideration at PLOS Computational Biology. As with all papers reviewed by the journal, your manuscript was reviewed by members of the editorial board and by several independent reviewers. The reviewers appreciated the attention to an important topic. Based on the reviews, we are likely to accept this manuscript for publication, providing that you modify the manuscript according to the review recommendations.

The reviewers were generally satisfied with your revisions of the manuscript. I would encourage you to consider the final remaining comments before a final decision can be made.

Sincerely,

Gunnar Blohm, Ph.D.

Associate Editor

PLOS Computational Biology

Samuel Gershman

Deputy Editor

PLOS Computational Biology

[LINK]

Reviewer's Responses to Questions

**Comments to the Authors:**

Reviewer #1: The authors have provided thoughtful responses to all of the questions I have previously raised. I also believe that the authors have done a great job in revising the manuscript to include additional explanation of their analytical frameworks, results, and discussion. The present manuscript addresses an important and timely topic which will have a wide-reaching impact in field of decision-making and human cognition.

I only have one further comment which I hope the authors could help clarify – in your response about t0. I’m not sure I necessarily agree with this statement: “In particular, if all trials were included, the non-decision time would necessarily adjust, shifting the modeled

RT distribution towards zero as much as required to ensure a probability density

> 0 for fast outlier trials.” Could the author please expand on this idea a bit more? Also, relatedly, I’m wondering if the authors think t0 could inform us something meaningful across experiments that use the same exact stimulus setup but with different response schemes (e.g., gaze shifting/grip force vs button press vs continuous dial response)?

Reviewer #2: The authors did a great job in their revision. I still think the paper could be clearer at the conceptual level, but the analyses are technically sound and the results certainly deserve publication. There is no need to prolong this review process with more iterations, but here is a couple of points that could be improved:

1) The paper is introduced with the notion of effort discounting (how a reward is devalued by the effort required to get it), which is not the question of interest in this paper. The authors justify this introduction by stating that their paper reveals another aspect of effort – in addition to reflecting a cost, it would reflect motivational process. But the two aspects are the two sides of the same coin: if motivation is measured by the cost one is willing to pay for a given reward, then accepting a higher cost (exerting more effort) reveals stronger motivation without changing the basic notion that effort is a cost. In other words, the literature on effort discounting also uses effort to reveal motivation, there is no novelty here. The difference, as stated later by the authors, is that effort is an implicit measure in the present study because it is not instrumental in getting a reward. The question of interest (whether implicit measures can provide further insight into subjective valuation during decision-making) without the misleading detour through effort discounting.

2) The interest of the approach is sold with the perspective of applications to neuropsychiatric diseases. I know everyone does that and I’m not particularly shocked, except by the justification that response vigor (grip force) measurement would be informative in conditions where effort processing is impaired (like Parkinson’s disease). To put it bluntly, this is like saying that eye-tracking would be helpful to investigate blind people because their visual exploration is impaired. The paper is interesting enough to attract attention without using this sort of twist.

Reviewer #3: I am happy with the changes

Reviewer #4: Thank you to the authors for their responses and additional analysis. I have a few additional comments:

1) Please report the times (mean, sd) observed between pre-selection and choice registration with the force transducer. This is useful information for future studies investigating similar questions. For example, one could ask how close does the time of the decision need to be to the motor response for this vigor effect to be observed. If the motor response is much delayed (say 1 min) would it still reflect the decision variables? Thus providing the numbers observed in this study will be helpful for others.

2) I am not clear what the second hypothesis is stating. As written, it says: "(ii) Subjective utility modulates response times and grip force: Faster response times and stronger effort (handgrip force) for choices with high utility (subjective value)."

It seems they mean ‘the difference in subjective utility’ rather than simply ‘subjective utility’? If not, then please clarify subjective utility of which option will modulate response times and grip force.

3) In section ‘3.4.2 Value sum effect’, please add a sentence at the end of the paragraph to summarize the main findings.

4) The results are a little difficult to follow. It would be helpful if the authors let the readers know when results supported their hypotheses (and specificy which hypothesis). Even better, would be a summary at the end of the results or in the discussion where there was a recap of the hypotheses and what was supported or rejected.

5) Can the authors clarify whether the new analysis of the effect of total value on response vigor hold true even within a condition? Is the effect they observe driven by the condition effect (high vs low), or is this something observed even with the low condition or high condition, independently?

**Have the authors made all data and (if applicable) computational code underlying the findings in their manuscript fully available?**

Reviewer #1: Yes

Reviewer #2: None

Reviewer #3: Yes

Reviewer #4: Yes

PLOS authors have the option to publish the peer review history of their article (what does this mean?). If published, this will include your full peer review and any attached files.

Reviewer #1: No

Reviewer #2: No

Reviewer #3: No

Reviewer #4: No

Figure Files:

Data Requirements:

Reproducibility:

References:

---

## [Editor Report · Decision Letter 2]

12 Apr 2022

Dear Dr. Smith,

We are pleased to inform you that your manuscript 'Motor response vigour and visual fixation patterns reflect subjective valuation during intertemporal choice' has been provisionally accepted for publication in PLOS Computational Biology.

Best regards,

Gunnar Blohm, Ph.D.

Associate Editor

PLOS Computational Biology

Samuel Gershman

Deputy Editor

PLOS Computational Biology

---

## [Editor Report · Acceptance letter]

26 May 2022

PCOMPBIOL-D-21-02032R2 

Motor response vigour and visual fixation patterns reflect subjective valuation during intertemporal choice

Dear Dr Smith,

I am pleased to inform you that your manuscript has been formally accepted for publication in PLOS Computational Biology. Your manuscript is now with our production department and you will be notified of the publication date in due course.

With kind regards,

Livia Horvath
